# From Decoupled to Coupled: Robustness Verification for Learning-based Keypoint Detection with Joint Specifications

## Abstract

Keypoint detection underpins many vision pipelines, from human-pose estimation and viewpoint recovery to 3D reconstruction. Yet, modern neural models remain vulnerable to subtle input variations. Despite its importance, robustness verification for keypoint detection remains largely unexplored due to the high dimensionality of input spaces and the complexity of deep models. In this work, we verify a property that bounds the joint deviation across all keypoints, capturing interdependencies among keypoints specified by system designers or derived from downstream performance requirements (e.g., pose-based error budgets). A few existing approaches reformulate the problem by decoupling each keypoint (or its neighboring pixels) into independent classification tasks, leading to overly conservative guarantees and fails to account for the collective role keypoints play in downstream tasks. We address this gap with the first coupled robustness verification framework for heatmap-based keypoint detectors under joint specifications. Our method supports any backbone architecture (e.g., CNN, ResNet, Transformer) that produces per-keypoint heatmaps, followed by a max-activation operation to extract coordinates. To do so, we combine the reachability and optimization methodologies by formulating robustness verification as a property falsification problem using a Mixed-Integer Linear Program (MILP) that combines (i) reachable sets of heatmap outputs, obtained via existing reachability analysis tools, and (ii) a polytope encoding the joint keypoint deviation constraint. Infeasibility of the MILP certifies robustness, while feasibility yields a potential counterexample. We prove that our method is sound, that is, it certifies robustness only when the property truly holds. Experiments demonstrate that our coupled method achieves a verified rate comparable to the testing-based method when the keypoint error thresholds are not tight. Moreover, under stricter keypoint error thresholds, our method maintains a high verified rate, whereas the decoupled approach fails to verify the robustness of any image in these scenarios.

## 1 Introduction

Keypoint detection is a fundamental computer vision problem that involves identifying distinctive locations or landmarks within an image. It serves as a critical building block for numerous downstream tasks in vision. For example, reliably detecting human body keypoints (e.g. joints) is the basis of pose estimation algorithms, enabling the interpretation of human posture from images (Sun et al., 2019a). In fact, accurate keypoint localization is central to a wide range of visual understanding applications – including viewpoint estimation (Zhou et al., 2018), action recognition (Hachiuma et al., 2023), feature matching (Sarlin et al., 2020), and 3D reconstruction (Novotny et al., 2022) – highlighting its broad importance. The success of these high-level tasks critically depends on the precision and reliability of the underlying keypoint detectors.

Nowadays, deep neural networks have emerged as the dominant paradigm for keypoint detection, owing to their ability to learn rich, task-specific feature representations from data. However, neural networks are known to be sensitive to distribution shifts and adversarial perturbations (Goodfellow et al., 2014), and keypoint detectors are no exception. Even subtle changes in an input image – such as occlusions, changes in lighting, or small noise patterns – can lead a neural network to predict incorrect or mislocalized keypoints. Researchers have proposed various training-time techniques to bolster the empirical robustness of keypoint

models (e.g. extensive data augmentation (Liu et al., 2020), adversarial training (Zhu et al., 2021), and specialized loss functions (Cheng et al., 2019) to handle occlusions or illumination changes). While these strategies can improve resilience to common perturbations, they do not guarantee reliability under all possible conditions. In particular, they provide no formal assurances that a trained keypoint detector will behave correctly on worst-case input variations that were not seen during training.

Robustness verification of keypoint detectors, proving that a model's outputs remain stable under specified input perturbations – remains largely unexplored. To date, most formal verification research in computer vision has focused on simpler tasks like image classification (Zhou et al., 2024; Brix et al., 2024), where the desired property (e.g. the predicted class label remains unchanged) is relatively straightforward to define and verify. In contrast, keypoint detection produces point coordinates, making it more challenging to formally define and verify robustness (since "approximately correct" keypoint locations must be tolerated rather than exact equality). As a result, very few works have tackled formal robustness guarantees for keypoint detection models (Kouvaros et al., 2023; Anonymity)[1]. These approaches typically reformulate the keypoint detection verification problem as a classification verification task by treating each keypoint—or pixels in its vicinity—as a distinct class. However, this decoupled approach overlooks the collective influence of keypoints on downstream tasks such as pose estimation, thereby leading to conservative results. This gap is especially concerning because keypoint detectors are increasingly deployed in safety-critical settings, such as autonomous driving, robotics, aerospace, and augmented reality, where prediction failures can have serious consequences.

In this work, we focus on a commonly used architecture for keypoint detection, where the input image is processed to generate a set of heatmaps—each representing the likelihood of a pixel being a specific keypoint. The keypoint locations are then extracted by applying a `max` activation operation over these heatmaps. The backbone network responsible for producing heatmaps can adopt various architectures, including convolutional, ResNet-based, or transformer-based designs. To bridge the gap in formal robustness verification, we target a property that captures the interdependency among keypoints—for example, requiring that the total deviation across all predicted keypoints remains below a specified threshold. Such a property can be specified either by system designers or derived from the requirements of downstream tasks. For instance, Anonymity translate pose error bounds into bounds on keypoint deviations. In contrast to prior works (Kouvaros et al., 2023; Anonymity), which treat each keypoint independently with its own error bound, our coupled approach preserves the collective behavior of keypoints without isolating them. Our main contribution is a MILP-based formulation that attempts to falsify this robustness property. This formulation integrates reachable sets of the heatmaps—obtained using various reachability analysis techniques—and a specification over joint keypoint deviations represented as a polytpe, as depicted in Fig. 1. If the MILP is infeasible, no counterexample exists that violates the property, certifying robustness. Conversely, if the MILP is feasible, we cannot guarantee robustness and the property remains unverified. Theoretically, we prove that our approach is sound, meaning it certifies robustness only when the specified property truly holds.

## 2 Related Work

### 2.1 Formal verification of neural networks

The objective of verifying neural networks involves ensuring they meet certain standards of safety, security, accuracy, or robustness. This essentially means determining the truth of a specific claim about the outputs of a network based on its inputs. In recent years, there has been a significant influx of research in this area. For comprehensive insights into neural network verification, one can refer to Liu et al. (2021). Verification techniques are generally divided into three main groups: reachability-based approaches, which perform a layer-by-layer analysis to assess network output range (Gehr et al., 2018; Xiang et al., 2018; Tran et al., 2020; Choi et al., 2025); optimization methods, which seek to disprove the assertion (Bastani et al., 2016; Tjeng et al., 2018; Banerjee et al., 2024); and search-based strategies which combine with reachability analysis or optimization to identify instances that contradict the assertion (Katz et al., 2019; Xu et al., 2020; Wu et al.,

---

[1] Due to the double-blind review process, we omit the details of this prior work and refer to it as "baseline" in the following text.

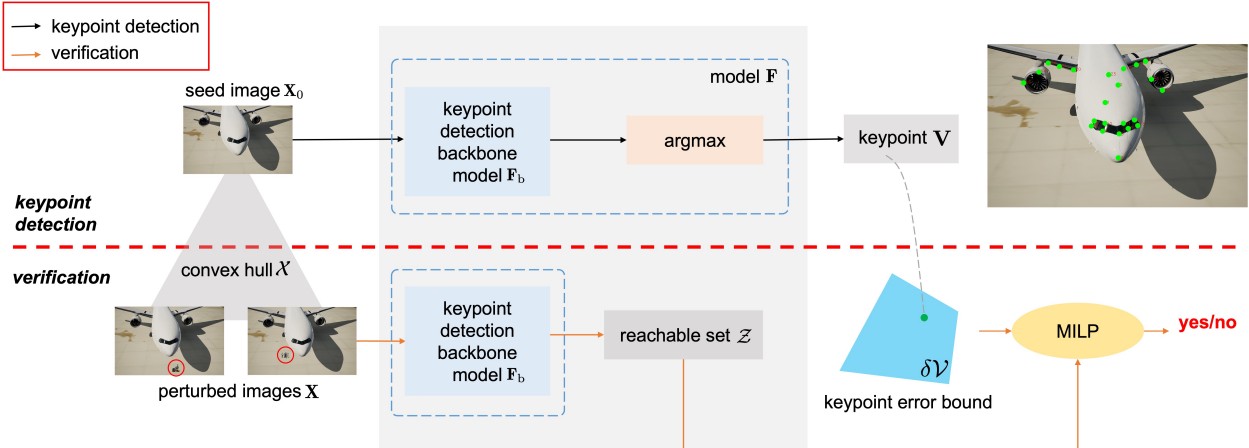

Figure 1: Overview of the keypoint detection pipeline and the proposed verification framework. A thick red dashed line divides the sections for keypoint detection (above) and verification (below). A seed image $\mathbf{X}_0$ where an airplane is parking at the airport is processed by the model to identify keypoints, which are marked as green dots. The verification framework takes as input the seed image $\mathbf{X}_0$ and a set of perturbed images $\mathbf{X}$ (with local perturbations indicated by red circles) that form the convex hull $\mathcal{X}$, along with the keypoint error bound $\delta\mathcal{V}$. By checking the feasibility of the MILP that is derived from the reachable set $\mathcal{Z}$ of the backbone model and the keypoint error bound $\delta\mathcal{V}$, the verification tool returns whether the model is robust.

2024; Duong et al., 2024). In 2020, VNN-COMP (Brix et al., 2023) launched as a competition to evaluate the capabilities of advanced verification tools spanning a variety of tasks, including collision detection, image classification, dataset indexing, and image generation. However, these methods treat deep neural networks in isolation, concentrating on analyzing the input-output relationship.

Concurrently, there is research focused on the system-level safety of closed-loop cyber-physical systems (CPS) incorporating neural network components, particularly within the system and controls domain. They broadly fall into two categories. The first category (Tran et al., 2019; Dutta et al., 2019; Everett, 2021; Ivanov et al., 2021a) focuses on ensuring the correctness of neural network-based controllers, taking their input from the structured outcomes of the state estimation module, regardless of whether the state estimation module is based on perception or not. Neural network controllers of this type generally consist of several fully connected layers, making them relatively straightforward to verify. The second category focuses on validating the closed-loop performance of vision-based dynamic systems that incorporate learning-based components. Among these, studies (Sun et al., 2019b; Ivanov et al., 2020; 2021b; Hsieh et al., 2022; Sun et al., 2022) examine LiDARs as the perception module, processed by multi-layer perceptrons (MLPs) with a few hidden layers. Other approaches, primarily applied to runway landing and lane tracking, deal with high-dimensional inputs from camera images, employing methods like approximate abstraction of the perception model (Hsieh et al., 2022), contract synthesis (Astorga et al., 2023), simplified networks within the perception model (Cheng et al., 2020; Katz et al., 2022), or a domain-specific model of the image formation process (Santa Cruz & Shoukry, 2022). Nevertheless, few studies directly handle camera-image inputs, owing to their unstructured, high-dimensional nature, in contrast with structured robot states such as position and velocity.

## 2.2 Certification of keypoint detection

The investigation of verification methods for keypoint detection is relatively limited. Talak et al. (2023) introduced a certifiable approach to keypoint-based pose estimation from point clouds by correcting keypoints identified by the model, ensuring the correctness guarantee of the pose estimation. Shi et al. (2023) expanded on this by integrating the correction concept with ensemble self-training. Similarly, by propagating the uncertainty in the keypoints to the object pose, Yang & Pavone (2023) created a keypoint-based pose estimator for point clouds that is provably correct and is characterized by definitive worst-case error bounds. While

these efforts focus on point clouds, in the image domain, Holmes et al. (2025) introduced an optimization layer for deep-learning networks that provides certifiably correct and differentiable solutions when the relaxation is tight, and demonstrated its application in detecting image keypoints for robot localization under challenging lighting conditions. Of all these studies, Kouvaros et al. (2023); Anonymity are the most closely related to our work, as they address the robustness verification of keypoint detection networks. However, in contrast to their decoupled verification approaches, we verify all keypoints jointly.

## 3 Background

In this work, we represent scalars and scalar functions by italicized lowercase letters ($x$), vectors and vector functions by upright bold lowercase letters ($\mathbf{x}$), matrices and matrix functions by upright bold uppercase letters ($\mathbf{X}$), and sets and set functions with calligraphic uppercase letters ($\mathcal{X}$). Let $\mathbb{R}$ and $\mathbb{Z}$ represent the sets of real and integer numbers, respectively.

### 3.1 Keypoint detection

A common strategy to detect keypoints involves the use of heatmap regression, wherein ground-truth heatmaps are created by placing 2D Gaussian kernels atop each keypoint. The heatmap pixel values are interpreted as the likelihood of each pixel being a keypoint. These heatmaps are then used to guide the training through an $\ell_2$ loss. The detection network can be divided into two parts. A *backbone* network, denoted by $\mathbf{F}_\mathrm{b}$, inputs a 2D image to produce heatmaps, one per keypoint, which is followed by an `argmax` operation for keypoint extraction. We refer to the argmax part as the *head* network. The entire network is represented by $\mathbf{V} = \mathbf{F}(\mathbf{X}) = \mathbf{F}_\mathrm{h} \circ \mathbf{F}_\mathrm{b}(\mathbf{X})$, where $\mathbf{X} \in \mathbb{R}^{H \times W \times C}$ represents a 2D RGB image with dimensions being $H \times W \times C$, and $\mathbf{V} \in \mathbb{Z}^{K \times 2}$ denotes the 2D coordinates of $K$ keypoints. Here, $\circ$ denotes function composition. In this paper, we impose no restrictions on the structure of the backbone model—it can be convolutional, ResNet-based, or transformer-based. To enhance accuracy and robustness, it is often essential to preprocess the input image $\mathbf{X}$ before it is passed to the network, such as resizing and color normalization. Denote this preprocessing step by $\mathbf{F}_0$, leading to the equation $\mathbf{V} = \mathbf{F}(\mathbf{X}) = \mathbf{F}_\mathrm{h} \circ \mathbf{F}_\mathrm{b} \circ \mathbf{F}_0(\mathbf{X})$. In what follows, we omit the preprocessing step unless it is critical to consider it.

### 3.2 Verification of neural networks

Consider a multi-layer neural network representing a function $\mathbf{f}$, which takes an input $\mathbf{x} \in \mathcal{D}_\mathbf{x} \subseteq \mathbb{R}^{d_0}$ and produces an output $\mathbf{y} \in \mathcal{D}_\mathbf{y} \subseteq \mathbb{R}^{d_n}$, where $d_0$ is the input dimension, and $d_n$ is the output dimension. Any non-vector inputs or outputs are restructured into vector form. The verification process entails assessing the validity of the following input-output relationships defined by the function $\mathbf{f}$: $\mathbf{x} \in \mathcal{X} \Rightarrow \mathbf{y} = \mathbf{f}(\mathbf{x}) \in \mathcal{Y}$, where sets, $\mathcal{X} \subseteq \mathcal{D}_\mathbf{x}$ and $\mathcal{Y} \subseteq \mathcal{D}_\mathbf{y}$, are referred to as input and output constraints, respectively.

In the context of confirming the robustness of a classification network, the goal is to ascertain that all samples within a proximal vicinity of a specified input $x_0$ receive an identical classification label. Assuming the target label is $i^* \in \{1, \ldots, d_n\}$, the specification for verification is that $y_{i^*} > y_j$ for every $j$ not equal to $i^*$. The constraints on inputs and outputs are established accordingly: $\mathcal{X} = \{\mathbf{x} \mid \|\mathbf{x} - \mathbf{x}_0\|_p \leq \epsilon\}, \mathcal{Y} = \{\mathbf{y} \mid y_{i^*} > y_j, \forall j \neq i^*\}$, where $\epsilon$ represents the maximum permissible deviation in the input space. The metric used to quantify disturbance can be any $\ell_p$ norm.

Neural network verification algorithms can generally be categorized into three main types: reachability analysis, optimization, and search. Neural network (NN) verification essentially seeks to transform the nonlinear model checking problem into piece-wise linear satisfiability problems, and it can be applied to various nonlinearities, including ReLU and, more recently, softmax (Wei et al., 2023). Two pivotal attributes—*soundness* and *completeness*—are of critical importance. A verification algorithm is *sound* if it only confirms the validity of a property when the property is indeed valid. It is *complete* if it consistently recognizes and asserts the existence of a property whenever it is actually present. There is a trade-off between computational complexity and conservativeness (or in-completeness).

## 4 Problem Formulation

In this work, we consider specifications that jointly constrain allowable deviations across keypoints, offering a more general formulation than prior work (Kouvaros et al., 2023; Anonymity) that introduces conservativeness by treating each keypoint independently. Our MILP-based method enables exact encoding of these joint specifications, thereby improving the overall completeness of the verification approach.

**Problem 1.** *Given a convex hull representation $\mathcal{X}$ consisting of a seed image $\mathbf{X}_0$ and $n$ perturbed images, defined by the set of all their possible convex combinations, i.e.,*

$$\mathcal{X} = \left\{ \mathbf{X} \;\middle|\; \mathbf{X} = \sum_{i=0}^{n} \omega_i \mathbf{X}_i, \quad s.t. \; \omega_i \geq 0, \sum_{i=0}^{n} \omega_i = 1 \right\}. \tag{1}$$

*Assume there are $K$ keypoints, and let $\delta\mathbf{v} \in \mathbb{Z}^{2K}$ denote the 2D keypoint error vector. The set of allowable keypoint deviations is represented by a polytope $\delta\mathcal{V} = \{\delta\mathbf{v} \,|\, \mathbf{P_v}\delta\mathbf{v} \leq \mathbf{b_v}\}$, where $\mathbf{P_v}$ and $\mathbf{b_v}$ are matrices of suitable dimensions. The goal is to certify whether the keypoint detection network $\mathbf{F} = \mathbf{F}_h \circ \mathbf{F}_b$ is robust to any image within the set $\mathcal{X}$. Mathematically,*

$$\forall \, \mathbf{X} \in \mathcal{X}, \; \mathbf{v} - \mathbf{v}^* \in \delta\mathcal{V} \quad s.t. \; \mathbf{v} = \mathbf{F}(\mathbf{X}), \tag{2}$$

*where $\mathbf{v}^* \in \mathbb{Z}^{2K}$ is the ground-truth coordinates of keypoints for the input image.*

**Remark 1.** *The permissible deviations in keypoint locations $\delta\mathcal{V}$, may be specified directly by design requirements or inferred from constraints imposed by downstream modules. Note that the allowable deviations among keypoints can exhibit interdependencies, as captured by the structure of the polytope $\delta\mathcal{V}$, which adds complexity to the problem. For example, Anonymity propagate system-level pose error bounds to keypoint-level bounds using sensitivity analysis. To make the verification tractable, their approach eliminates interdependencies among keypoints by identifying the largest axis-aligned hyper-rectangle within $\delta\mathcal{V}$, which dramatically reduces the allowable keypoint error bounds and introduces conservativeness.*

## 5 Robustness as Feasibility of the MILP

In addressing Problem 1, we identify two main challenges. First, keypoint detection operates over two spaces: the heatmap space and the coordinate space, linked through the process of maximum value extraction. Under perturbations, any valid heatmap must yield a maximum whose location lies within the allowable coordinate set. Second, the allowable deviations of keypoints are interdependent. Our approach addresses both challenges simultaneously by formulating a MILP that attempts to falsify the condition, i.e., to identify a heatmap within the reachable set whose maximum falls outside the allowable coordinate set defined by the joint output specifications.

Let $\mathcal{Z}$ denote the over-approximation of the reachable set of the backbone model, which can be obtained by various reachability analysis methods. Without loss of generality, we assume that this reachable set is a zonotope—a special type of convex polytope with a compact representation. In its matrix form, a zonotope is defined by a center $\mathbf{C} \in \mathbb{R}^{HW \times K}$ and a linear combination of a set of $m$ generators $\mathbf{G} \in \mathbb{R}^{HW \times K \times m}$. Mathematically, it is expressed as:

$$\mathcal{Z} = \left\{ \mathbf{Z} \in \mathbb{R}^{HW \times K} \;\middle|\; \mathbf{Z} = \mathbf{C} + \sum_{k=1}^{m} \alpha_k \mathbf{G}_k, \, \alpha_k \in [-1, 1] \right\}. \tag{3}$$

where $\mathbf{G}_k$ is the $k$-th generator, and each heatmap is flattened into a vector of length $HW$. Any point $\mathbf{Z}$ within the reachable set $\mathcal{Z}$ represents multi-channel heatmaps, which are used to predict keypoints by extracting the maximum value from each channel. The resulting keypoint errors are classified as *in-bound* if they fall within the polytope $\delta\mathcal{V}$. Otherwise, they are categorized as *out-of-bound*.

We formulate a MILP aiming to determine whether a point $\mathbf{Z} \in \mathcal{Z}$, i.e., multi-channel heatmaps, within the reachable set exists such that the corresponding keypoint errors are out of bound. If no such point exists, i.e.,

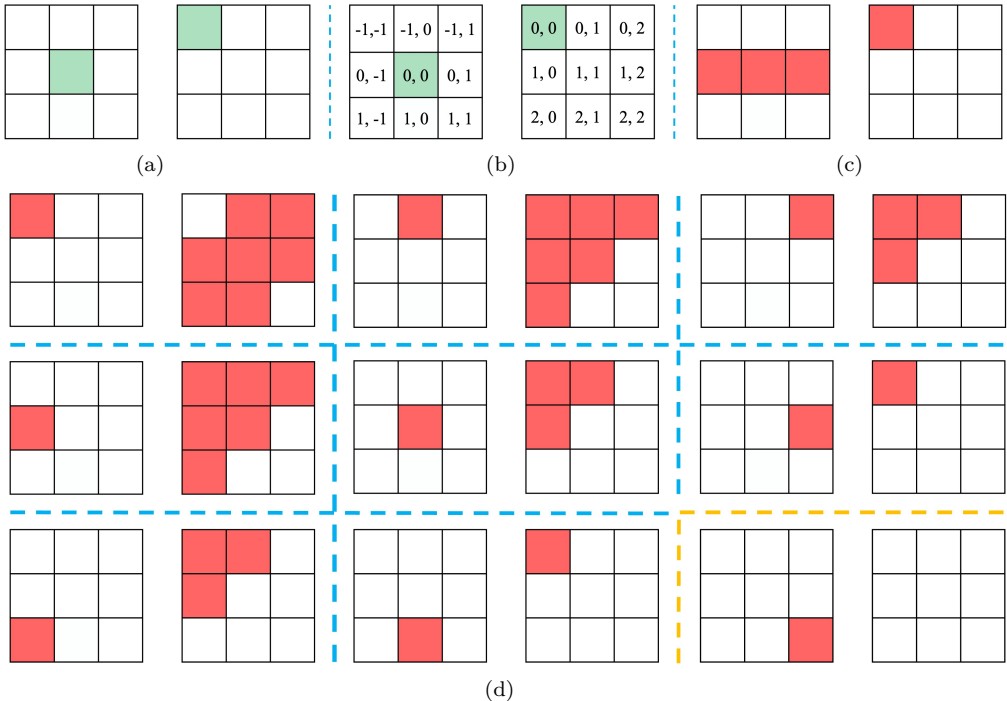

Figure 2: (a) Ground-truth keypoints are shown in green. (b) Deviations from the ground-truth keypoints are illustrated in both the vertical (first element) and horizontal (second element) directions. (c) Decoupled keypoint errors as determined by the method in the baseline (Anonymity). (d) Eight scenarios of coupled keypoint errors are evaluated using our approach, where the error of the first keypoint is fixed and the feasible error combinations for the second keypoint are enumerated. For example, the top-left case shows the feasible $\delta h_2$ and $\delta w_2$ values when $\delta h_1 = \delta w_1 = -1$. The bottom-right case illustrates cases where keypoint errors fall outside the polytope. Specifically, when the first keypoint deviates from the ground truth by 1 pixel in both directions, no feasible solution exists, even if the second keypoint aligns perfectly with the ground truth. Note that (c) and (d) are best interpreted with reference to (b) to evaluate satisfaction of the constraints in (4).

the MILP is infeasible, it indicates that the keypoint detection is considered robust. Conversely, if out-of-bound keypoint errors are identified, the robustness result is deemed unknown since typically the reachable set $\mathcal{Z}$ overapproximates the true reachable set. To this end, in Section 5.1, we formulate constraints that encode whether a point $\mathbf{Z}$ lies within the reachable set $\mathcal{Z}$ and whether a keypoint deviation $\delta\mathbf{v}$ violates the allowable keypoint deviations $\delta\mathcal{V}$. In Section 5.2, we further extract the pixel values of $\mathbf{Z}$ at the coordinates determined by the out-of-bound deviation $\delta\mathbf{v}$. If these pixels correspond to the maximal values in their respective channels, we identify a counterexample $\mathbf{Z}$, as the locations of these maximal activations yield keypoints outside the allowable bounds. We also propose strategies in Section 5.3 to reduce the size of the resulting MILP formulation to improve computational efficiency. Example 1 serves as a running example to illustrate the proposed approach.

**Example 1.** *We illustrate the advantages of our approach using a simple example with a $3 \times 3$ input image and two keypoints. In contrast to the "decoupled" method in Anonymity, which enforces independence among keypoint errors by disentangling them, our method preserves their interdependence. Assume the ground truth positions of the keypoints are as follows: the first keypoint is located at the center of the image, while the second keypoint is at the top-left corner (see Fig. 2(a)). Let $\delta h_i$ and $\delta w_i$, for $i = 1, 2$, represent the errors in the vertical and horizontal directions for the keypoints. The polytope $\delta\mathcal{V}$ representing the keypoint error*

*bound is defined as:*

$$\delta h_1 + \delta w_1 + \delta h_2 + \delta w_2 \le 1, \tag{4a}$$
$$\delta h_1 + \delta w_1 + \delta h_2 + \delta w_2 \ge -1. \tag{4b}$$

*Fig. 2(c) illustrates the independent keypoint error model from the baseline (Anonymity), where the allowable deviations are represented as red rectangles centered at the ground-truth positions, that is,*

$$\delta h_1 = 0, -1 \le \delta w_1 \le 1, \delta h_2 = 0, \delta w_2 = 0.$$

*Under this method, the first keypoint is permitted to deviate by one pixel in the horizontal direction but none vertically, while the second keypoint is not allowed any deviation. In contrast, our coupled approach leverages the entire feasible region within the polytope (4), as illustrated by the irregular geometry in Fig. 2(d). Clearly, the decoupled method in Anonymity is overly conservative, as it excludes a significant portion of feasible cases inside $\delta\mathcal{V}$. The bottom-right section of Figure 2(d) illustrates cases where the keypoint errors fall outside the allowable polytope $\delta\mathcal{V}$ (e.g., $\delta h_1 = 1$, $\delta w_1 = 1$).*

### 5.1 A point inside the reachable set and out-of-bound keypoint errors

Let $\mathcal{S} = \{1, \ldots, HW\}$ represent the set of flattened indices for each heatmap, and let $\mathbf{Z} \in \mathbb{R}^{HW \times K}$ denote the matrix of continuous variables. $\mathbf{Z}$ is contained in the zonotope if there exists a set of $\alpha_k \in [-1, 1]$ such that

$$\mathbf{Z} = \mathbf{C} + \sum_{k=1}^{m} \alpha_k \mathbf{G}_k. \tag{5}$$

For the $i$-th keypoint, let $(v_{2i-1}^*, v_{2i}^*) \in \mathbb{Z}^2$ denote the ground-truth 2D coordinates. The condition that $i$-th keypoint's deviation $(\delta v_{2i-1}, \delta v_{2i})$ from its true position remains within the image boundaries can be expressed as:

$$-(v_{2i-1}^* - 1) \ \le\ \delta v_{2i-1} \ \le\ H - v_{2i-1}^*, \quad \text{for } i = 1, \ldots, K, \tag{6a}$$
$$-(v_{2i}^* - 1) \ \le\ \delta v_{2i} \ \ \ \le\ W - v_{2i}^*, \quad \text{for } i = 1, \ldots, K. \tag{6b}$$

Define $\epsilon$ and $M$ as very small ($10^{-6}$) and very large ($10^6$) positive constants, respectively. Additionally, let $\mathbf{r} \in \{0, 1\}^r$ be a binary vector of size $r$, equal to the number of rows in the matrix $\mathbf{P_v}$. Using the Big-M method, the condition that $\delta\mathbf{v}$ lies outside the polytope of keypoint errors $\delta\mathcal{V}$ can be encoded as follows:

$$\sum_{i=1}^{r} r_i \ge 1, \tag{7a}$$
$$\mathbf{P_v} \delta\mathbf{v} \ge b_\mathbf{v} + \boldsymbol{\epsilon} - M(\mathbf{1} - \mathbf{r}). \tag{7b}$$

Here, $\boldsymbol{\epsilon}$ denotes a vector with all entries equal to $\epsilon$, and $r_i = 1$ indicates that the point $\delta\mathbf{v}$ lies outside the $i$-th face of the polytope $\delta\mathcal{V}$. To tighten constraint (7b), define $M_i$ as the element-wise maximum value of the vector $-\mathbf{P_v}\delta\mathbf{v} + b_\mathbf{v} + \boldsymbol{\epsilon}$, that is,

$$M_i \ = \ \max_{\delta\mathbf{v} \text{ s.t. } (6)} \left[ -\mathbf{P_v} \delta\mathbf{v} + b_\mathbf{v} + \boldsymbol{\epsilon} \right]_i,$$

where $[\cdot]_i$ extracts the $i$-th component. Assemble these values into the diagonal matrix $\mathbf{M} = \text{diag}(M_1, \ldots, M_r)$. Now constraint (7b) can be rewritten as

$$\mathbf{P_v} \delta\mathbf{v} \ \ge\ b_\mathbf{v} + \boldsymbol{\epsilon} - \mathbf{M}(\mathbf{1} - \mathbf{r}).$$

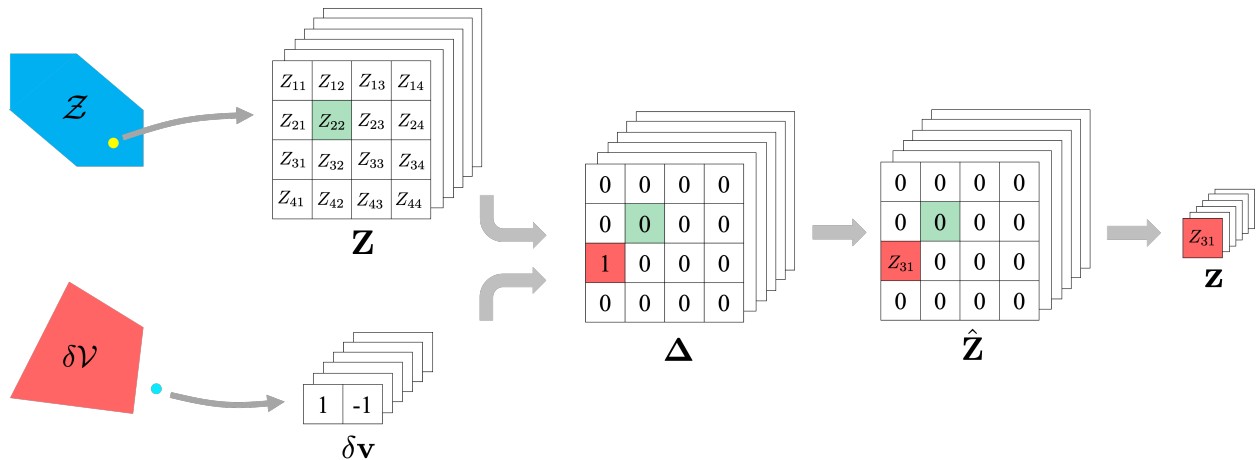

Figure 3: An illustration of dynamic indexing for the first keypoint: the green grid indicates the ground-truth keypoint location, while the red grid represents the perturbed keypoint position. The index deviation (1, -1) retrieves the value $Z_{31}$.

### 5.2 Dynamic indexing and and maximality check under keypoint deviations

Given a keypoint deviation vector $\delta \mathbf{v}$, this section describes the procedure for retrieving values from a point $\mathbf{Z} \in \mathcal{Z}$ using the index $\mathbf{v}_0 + \delta \mathbf{v}$, and verifying whether the corresponding projected values of $\mathbf{Z}$ are element-wise maximal. Since $\delta \mathbf{v}$ is itself composed of variables, we refer to this as the dynamic indexing problem, as depicted in Fig. 3.

#### 5.2.1 Binary indicator of $\mathbf{Z}$ at indexed location $(\mathbf{v}_0 + \delta \mathbf{v})$

Let $\boldsymbol{\Delta} \in \{0,1\}^{HW \times K}$, with the same dimensions as $\mathbf{Z}$, denote a matrix of binary variables, where the position indexed by $(\mathbf{v}_0 + \delta \mathbf{v})$ is set to 1, and all others are set to 0. The perturbed coordinates of the $i$-th keypoint are expressed as $(h_i, w_i) = (v^*_{2i-1} + \delta v_{2i-1}, v^*_{2i} + \delta v_{2i})$. For the $i$-th keypoint, $\boldsymbol{\Delta}$ being the binary indicator of $\mathbf{Z}$ is formulated as:

$$\sum_{j \in \mathcal{S}} \Delta_{ji} = 1, \tag{8a}$$

$$(h_i - 1)W + w_i = \sum_{j \in \mathcal{S}} j \Delta_{ji}. \tag{8b}$$

#### 5.2.2 Extracting values indexed by $(\mathbf{v}_0 + \delta \mathbf{v})$

Define $\underline{\mathbf{M}} \in \mathbb{R}^{HW \times K}$ and $\overline{\mathbf{M}} \in \mathbb{R}^{HW \times K}$ as the element-wise lower and upper bounds of the point $\mathbf{Z}$ within the zonotope $\mathcal{Z}$, respectively. Specifically, the lower bound is calculated by subtracting the absolute sum of contributions from all generators from the center value:

$$\underline{M}_{ji} = C_{ji} - \sum_{k=1}^{m} |G_{jik}|.$$

The upper bound is determined by adding the absolute sum of contributions from all generators to the center value:

$$\overline{M}_{ji} = C_{ji} + \sum_{k=1}^{m} |G_{jik}|.$$

Let $\hat{\mathbf{Z}} \in \mathbb{R}^{HW \times K}$ be a matrix of continuous variables where each element equals the corresponding element in $\mathbf{Z}$ if the $\boldsymbol{\Delta}$ value at the same location is 1, and is zero otherwise. This relationship is expressed as follows

for $j \in \mathcal{S}$:

$$\hat{Z}_{ji} \geq \underline{M}_{ji}\Delta_{ji}, \tag{9a}$$

$$\hat{Z}_{ji} \leq \overline{M}_{ji}\Delta_{ji}, \tag{9b}$$

$$\hat{Z}_{ji} \leq Z_{ji} - \underline{M}_{ji}(1 - \Delta_{ji}), \tag{9c}$$

$$\hat{Z}_{ji} \geq Z_{ji} - \overline{M}_{ji}(1 - \Delta_{ji}). \tag{9d}$$

If $\Delta_{ji} = 1$, constraints (9c) and (9d) ensure that $\hat{Z}_{ji} = Z_{ji}$. Conversely, if $\Delta_{ji} = 0$, constraints (9a) and (9b) enforce that $\hat{Z}_{ji} = 0$. Let $\mathbf{z} \in \mathbb{R}^K$ represent the set of continuous variables corresponding to values in $\mathbf{Z}$ indexed by the perturbed coordinates $\mathbf{v}^* + \delta\mathbf{v}$. These values are computed as:

$$z_i = \sum_{j \in \mathcal{S}} \hat{Z}_{ji}. \tag{10}$$

The feasible set of the variable $\mathbf{z}$, defined by constraints (7) through (10), corresponds to a lower-dimensional projection of the reachable set, determined by out-of-bound keypoint errors. We refer to this as the *out-of-bound projection*, whereas projections associated with admissible keypoint deviations are termed *in-bound projections*.

### 5.2.3 Finding a counterexample

Let $\hat{\mathcal{Z}}_{\text{in}} \subseteq \mathbb{R}^K$ represent the sets of in-bound projections. The objective is to determine whether there exists a $\mathbf{z}$ such that it is element-wise larger than any point $\mathbf{z}_{\text{in}} \in \hat{\mathcal{Z}}_{\text{in}}$. Focusing on the $i$-th keypoint, this means the $i$-th component $z_i$ must exceed all possible values of $\mathbf{z}_{\text{in}}^i$. To compute all possible values of $\mathbf{z}_{\text{in}}^i$, we project the polytope of keypoint errors $\delta\mathcal{V}$ onto the $i$-th keypoint, denoting the resulting projected polytope as $\delta\mathcal{V}_i$. By enumerating all pairs of integer values $(\delta v_{2i-1}, \delta v_{2i})$ within this polytope, we obtain the set of in-bound keypoint errors for the $i$-th keypoint. These integer pairs must also satisfy the image boundaries constraint (6). Each pair of in-bound errors is then added to the ground-truth coordinate $(v_{2i-1}^*, v_{2i}^*)$ to compute the perturbed coordinates, and we denote the resulting set of flattened in-bound indices as $\mathcal{S}_{\text{in}}^i$. Finally, the condition that $z_i$ is greater than all possible values of $\mathbf{z}_{\text{in}}^i$ can be expressed as:

$$z_i \geq Z_{ji}, \quad \forall j \in \mathcal{S}_{\text{in}}^i. \tag{11}$$

### 5.3 Pruning the size of MILP

Constraint (11) ensures that $z_i$ exceeds the largest value within the set $\mathcal{S}_{\text{in}}^i$. To reduce the number of variables in constraint (11), we aim to minimize the cardinality of $\mathcal{S}_{\text{in}}^i$ by eliminating flattened indices for which another flattened index has a minimum value exceeding its maximum value. This is formally defined as:

$$\mathcal{S}_{\text{in}}^{i,-} = \left\{ j \in \mathcal{S}_{\text{in}}^i \mid \exists j' \in \mathcal{S}_{\text{in}}^i \text{ and } j' \neq j, \ \underline{M}_{j'i} \geq \overline{M}_{ji} \right\}. \tag{12}$$

We then define a refined set of in-bound indices as $\mathcal{S}_{\text{in}}^{i,*} = \mathcal{S}_{\text{in}}^i \setminus \mathcal{S}_{\text{in}}^{i,-}$, replacing $\mathcal{S}_{\text{in}}^i$ with $\mathcal{S}_{\text{in}}^{i,*}$ in constraint (11).

Similarly, constraints (8)-(10) relate to the out-of-bound indices. The candidate set of out-of-bound indices is defined as $\mathcal{S}_{\text{out}}^i = \mathcal{S} \setminus \mathcal{S}_{\text{in}}^{i,-}$. From this set, we further remove indices where an in-bound index exists such that its minimum value is no less than the maximum value of the out-of-bound index. This is described as:

$$\mathcal{S}_{\text{out}}^{i,-} = \left\{ j \in \mathcal{S}_{\text{out}}^i \mid \exists j' \in \mathcal{S}_{\text{in}}^i \text{ and } j' \neq j, \ \underline{M}_{j'i} \geq \overline{M}_{ji} \right\}. \tag{13}$$

The pruned set of out-of-bound indices is then defined as $\mathcal{S}_{\text{out}}^{i,*} = \mathcal{S}_{\text{out}}^i \setminus \mathcal{S}_{\text{out}}^{i,-}$. Finally, we refine constraint (5), (8), (9) and (10) by replacing $\mathcal{S}$ with $\mathcal{S}_{\text{in}}^{i,*} \cup \mathcal{S}_{\text{out}}^{i,*}$, as the original $\mathcal{S}$ encompasses both in-bound and out-of-bound indices.

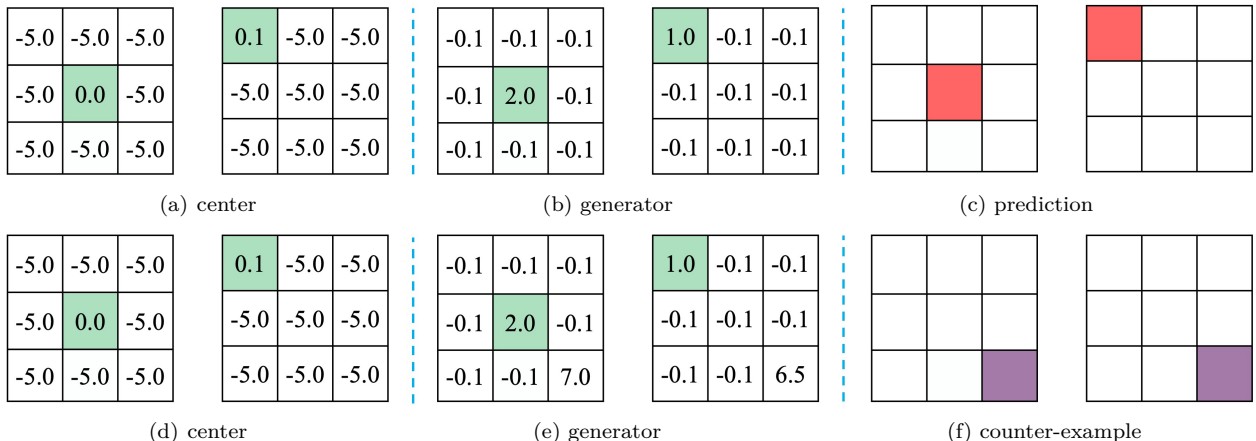

Figure 4: In the first scenario (a)-(c), the MILP is infeasible. Conversely, in the second scenario (d)-(f), the MILP is feasible, resulting in the identification of a counterexample.

**Example 1.** *(continued) We analyze a reachable set defined by a center and a single generator, considering two scenarios where the MILP is infeasible in the first scenario and feasible in the second. In the first scenario, the center* $\mathbf{C}$ *and generator* $\mathbf{G}$ *are depicted in Fig. 4(a) and Fig. 4(b), respectively. For any point in the reachable set* $\mathbf{C} + \alpha\mathbf{G}$ *with* $\alpha$ *ranging from -1 to 1, the minimum values at the ground-truth pixels in each heatmap are higher than the maximum values at all other pixels. Consequently, the predicted keypoints align perfectly with the ground truth (as shown in Fig. 4(c)). Therefore, the MILP is infeasible because there are no out-of-bound locations whose values exceed those of the in-bound locations. In the second scenario, using a different generator, the maximum values in the heatmaps occur at the bottom-right pixels when* $\alpha = 1$ *(see Fig. 4(f)). This results in keypoint errors of* $\delta h_1 = 1, \delta w_1 = 1$*, and* $\delta h_2 = 2, \delta w_2 = 2$ *(refer to Fig. 2(b)), which violate the constraints defined by (4). As a result, a counterexample is identified, where keypoint errors fall outside the polytope* $\delta\mathcal{V}$*. We present the MILP formulation for the first scenario. First, the zonotope in (5) is expressed as*

$$\mathbf{Z} = \begin{bmatrix} -5.0 & -5.0 & -5.0 & -5.0 & 0.0 & -5.0 & -5.0 & -5.0 & -5.0 \\ 0.1 & -5.0 & -5.0 & -5.0 & -5.0 & -5.0 & -5.0 & -5.0 & -5.0 \end{bmatrix}^{\mathsf{T}}$$
$$+ \alpha_1 \begin{bmatrix} -0.1 & -0.1 & -0.1 & -0.1 & 2.0 & -0.1 & -0.1 & -0.1 & -0.1 \\ 1.0 & -0.1 & -0.1 & -0.1 & -0.1 & -0.1 & -0.1 & -0.1 & -0.1 \end{bmatrix}^{\mathsf{T}}, \quad \alpha_1 \in [-1, 1].$$
(14)

*Let* $\delta\mathbf{v} = [\delta v_1, \delta v_2, \delta v_3, \delta v_4]$*. Given that the heatmap has dimensions* $H = 3$ *and* $W = 3$*, and the ground-truth coordinates are* $(v_1^*, v_2^*) = (2, 2)$ *and* $(v_3^*, v_4^*) = (1, 1)$*, the boundary condition for* $\delta\mathbf{v}$ *in constraint (6) can be expressed as*

$$-1 \leq \delta v_1 \leq 1, \quad -1 \leq \delta v_2 \leq 1, \tag{15a}$$
$$0 \leq \delta v_3 \leq 2, \quad 0 \leq \delta v_4 \leq 2. \tag{15b}$$

*Let* $\mathbf{r} = [r_1, r_2]$ *and* $\epsilon = 1 \times 10^{-6}$*. The element-wise maximum values of the two entries in the vector* $-\mathbf{P_v}\delta\mathbf{v} + b_\mathbf{v} + \epsilon$ *are 3.000001 and 7.000001, respectively. Accordingly, the out-of-polytope condition (7) for* $\delta\mathcal{V}$ *can be written as*

$$r_1 + r_2 \geq 1, \tag{16a}$$
$$\begin{bmatrix} 1 & 1 & 1 & 1 \\ -1 & -1 & -1 & -1 \end{bmatrix} \delta\mathbf{v} \geq \begin{bmatrix} -2 + 3.000001r_1 \\ -6 + 7.000001r_2 \end{bmatrix}. \tag{16b}$$

The perturbed coordinates are $(h_1, w_1) = (2+\delta v_1, 2+\delta v_2)$ and $(h_2, w_2) = (1+\delta v_3, 1+\delta v_4)$. Let $\mathcal{S} = \{1, \ldots, 9\}$ and $\boldsymbol{\Delta} \in \{0,1\}^{9 \times 2}$. The binary indicator constraint (8) can then be written as

$$1 = \sum_{j \in \mathcal{S}} \Delta_{j1}, \quad 3\delta v_1 + \delta v_2 + 5 = \sum_{j \in \mathcal{S}} j \Delta_{j1}, \tag{17a}$$

$$1 = \sum_{j \in \mathcal{S}} \Delta_{j2}, \quad 3\delta v_3 + \delta v_4 + 1 = \sum_{j \in \mathcal{S}} j \Delta_{j2}. \tag{17b}$$

The element-wise lower and upper bounds for the points contained in the zonotope $\mathcal{Z}$ are

$$\underline{\mathbf{M}} = \begin{bmatrix} -5.1 & -5.1 & -5.1 & -5.1 & -2.0 & -5.1 & -5.1 & -5.1 & -5.1 \\ -0.9 & -5.1 & -5.1 & -5.1 & -5.1 & -5.1 & -5.1 & -5.1 & -5.1 \end{bmatrix}^{\mathsf{T}},$$

$$\overline{\mathbf{M}} = \begin{bmatrix} -4.9 & -4.9 & -4.9 & -4.9 & 2.0 & -4.9 & -4.9 & -4.9 & -4.9 \\ 1.1 & -4.9 & -4.9 & -4.9 & -4.9 & -4.9 & -4.9 & -4.9 & -4.9 \end{bmatrix}^{\mathsf{T}}.$$

Therefore, constraint (9) becomes

$$\hat{\mathbf{Z}} \geq \underline{\mathbf{M}} \odot \boldsymbol{\Delta}, \tag{18a}$$

$$\hat{\mathbf{Z}} \leq \overline{\mathbf{M}} \odot \boldsymbol{\Delta}, \tag{18b}$$

$$\hat{\mathbf{Z}} \leq \mathbf{Z} - \underline{\mathbf{M}} \odot (\mathbf{1} - \boldsymbol{\Delta}), \tag{18c}$$

$$\hat{\mathbf{Z}} \geq \mathbf{Z} - \overline{\mathbf{M}} \odot (\mathbf{1} - \boldsymbol{\Delta}). \tag{18d}$$

where $\odot$ denotes element-wise multiplication. Let $\mathbf{z} = [z_1, z_2]$. Then, the summation constraint (10) can be expressed as

$$z_1 = \sum_{j \in \mathcal{S}} \hat{Z}_{j1}, \quad z_2 = \sum_{j \in \mathcal{S}} \hat{Z}_{j2}. \tag{19}$$

The projection of $\delta \mathcal{V}$ onto the first keypoint yields $\{(-1,-1), (-1,0), (-1,1), (0,-1), (0,0), (0,1), (1,-1), (1,0)\}$. Given the ground-truth coordinate $(2,2)$, this corresponds to the 2D coordinates $\{(1,1), (1,2), (1,3), (2,1), (2,2), (2,3), (3,1), (3,2)\}$. The corresponding flattened in-bound indices for the first keypoint are $\mathcal{S}_{\text{in}}^1 = \{1, \ldots, 8\}$. Similarly, $\mathcal{S}_{\text{in}}^2 = \{1, \ldots, 8\}$. Therefore, the counterexample constraint (11) becomes

$$z_1 \geq Z_{j1}, \quad j \in \mathcal{S}_{\text{in}}^1, \tag{20a}$$

$$z_2 \geq Z_{j2}, \quad j \in \mathcal{S}_{\text{in}}^2. \tag{20b}$$

For the first keypoint, the lower bound (-2.0) of the value at index 5 in the first column of $\underline{\mathbf{M}}$ exceeds the upper bound (-4.9) of all other values in the first column of $\overline{\mathbf{M}}$, giving $\mathcal{S}_{\text{in}}^{1,-} = \{1,2,3,4,6,7,8\}$. Consequently, $\mathcal{S}_{\text{in}}^{1,*} = \mathcal{S}_{\text{in}}^1 \setminus \mathcal{S}_{\text{in}}^{1,-} = \{5\}$. For the set of out-of-bound indices, we have $\mathcal{S}_{\text{out}}^1 = \mathcal{S} \setminus \mathcal{S}_{\text{in}}^{1,-} = \{5,9\}$, $\mathcal{S}_{\text{out}}^{1,-} = \{9\}$. Hence, $\mathcal{S}_{\text{out}}^{1,*} = \mathcal{S}_{\text{out}}^1 \setminus \mathcal{S}_{\text{out}}^{1,-} = \{5\}$. Similarly, for the second keypoint, $\mathcal{S}_{\text{in}}^{2,*} = \{1\}, \mathcal{S}_{\text{out}}^{2,*} = \{1\}$. We found that

$$\mathcal{S}_{\text{in}}^{1,*} = \mathcal{S}_{\text{out}}^{1,*} = \{5\}, \quad \mathcal{S}_{\text{in}}^{2,*} = \mathcal{S}_{\text{out}}^{2,*} = \{1\}$$

which correspond exactly to the locations of the ground-truth pixels. In other words, for each keypoint, the pruned set of out-of-bound indices coincides with the pruned set of in-bound indices. This renders the formulation for the scenario infeasible, since an index cannot simultaneously be both in-bound and out-of-bound.

Following the same logic, for the second scenario in Figure 4, we obtain

$$\mathcal{S}_{\text{in}}^{1,*} = \{5\}, \quad \mathcal{S}_{\text{out}}^{1,*} = \{5,9\}, \quad \mathcal{S}_{\text{in}}^{2,*} = \{1\}, \quad \mathcal{S}_{\text{out}}^{2,*} = \{1,9\}.$$

In this case, the MILP formulation has a feasible solution with $\alpha_1 = 1.0, \delta\mathbf{v} = [1,1,2,2], \mathbf{r} = [1,0], \Delta_{91} = \Delta_{92} = 1, z_1 = Z_{91}, z_2 = Z_{92}$.

# 6 Theoretical Analysis

In this section, we present the theoretical foundation of our verification framework. We formally establish the soundness of the proposed MILP formulation.

**Theorem 1** (Soundness). *Our approach is sound: if it certifies the model as robust, then the keypoint detection model is guaranteed to be robust.*

*Proof.* The claim follows directly from the MILP formulation. Infeasibility of the MILP implies that there exists no point $\mathbf{Z} \in \mathcal{Z}$ for which the maximum value in each heatmap results in out-of-bound indices. Equivalently, every point in $\mathcal{Z}$ yields keypoint deviations that remain within the acceptable output set $\delta \mathcal{V}$. Since $\mathcal{Z}$ is an over-approximation of the model's true reachable set, we conclude that the model is robust. □

**Remark 2.** *Note that if our approach fails to certify robustness, it does not necessarily mean that the model is non-robust to the given perturbation. The failure may result from the conservativeness of the reachable set over-approximation used in the verification process.*

# 7 Evaluation Results

In this section, we empirically evaluate the proposed verification framework on a realistic keypoint-based airplane pose estimation task. We first describe the case study, including the dataset, backbone model, perturbation models, and output specifications, and then introduce the metrics used to assess both computational efficiency and verification accuracy. We subsequently present and analyze the results, comparing our method with the baseline and examining the influence of different perturbation types, overlap patterns, and specification tightness on the efficiency and accuracy.

## 7.1 Case study

### 7.1.1 Dataset

We adopt the keypoint-based pose estimation task presented in the baseline (Anonymity), which aims to estimate the pose of an airplane parked at an airport relative to a camera. The estimation pipeline follows a two-stage process: first, keypoints are detected in the input image; then, a Perspective-n-Point (PnP)-based nonlinear optimization algorithm computes the pose by leveraging known 3D-2D correspondences of keypoints. In this case, 23 keypoints are strategically placed across the airplane's surface to provide comprehensive coverage of the aircraft's geometry (marked as green dots in Fig. 1). The dataset comprises 7320 RGB images of airplanes, each with a resolution of $1920 \times 1200$ pixels.

### 7.1.2 Backbone model

The CNN-based model consists of five convolutional layers followed by five deconvolutional layers, designed to process input images of size $64 \times 64 \times 3$. In total, the network comprises 39 layers and contains approximately $6.57 \times 10^5$ trainable parameters.

### 7.1.3 Perturbations

From a dataset of over 7000 images, we randomly select 200 seed images and apply either local or global perturbations to generate the evaluation set.

**Local object occlusions**  We constructed a set of 40 realistic semantic disturbances representing common airport elements such as personnel and vehicles. To generate perturbed images, we randomly selected 20 objects from this set to serve as patches, each up to 150 pixels in size, and randomly placed one patch per image on the seed images (local objects are highlighted with red circles in Fig. 1). Perturbations were categorized as overlapping or non-overlapping, depending on whether the added patch overlaps with any part of the airplane in the image. Note that each convex hull used in analysis contains only overlapping or only non-overlapping images.

In total, we generated 4000 perturbed images: 893 classified as overlapping and 3107 as non-overlapping. To evaluate the impact of semantic disturbances on robustness, we varied the number of perturbed images from 1 to 4 to form the convex hull $\mathcal{X}$. These images were randomly sampled, enabling a controlled study of how increasing semantic complexity influences system performance and robustness guarantees.

**Global perturbations** We change each pixel's value through two types of global perturbations: brightness and contrast. For brightness, a variation value $b \in \mathbb{Z}$ is applied such that each pixel's value increases by $b$, that is, $I' = \texttt{clip}(I + b)$, where $I$ represents the original pixel values, $I'$ the new pixel values, and $\texttt{clip}$ ensures the values remain within the range $[0, 255]$. For contrast perturbation, a variation value $c \in \mathbb{R}$ adjusts each pixel's value by a percentage $c$, formulated as $I' = \texttt{clip}(I \times (1 + c))$. The convex hull is constructed as follows. For a positive value $b \in \mathbb{Z}_+$, two perturbed images are created for $b$ and $-b$, respectively. These images act as vertices of the convex hull. Along with the seed image, this approach facilitates verification of the model's robustness to any brightness variation within the range $[-b, b]$. The same methodology applies to contrast variations $c \in \mathbb{R}_+$. We examine the effects for $b$ values of $\{1, 2\}$ and $c$ values of $\{5 \times 10^{-4}, 5 \times 10^{-3}, 1 \times 10^{-2}\}$.

### 7.1.4 Output specification

The system requirement is specified in terms of pose error thresholds, given by $\boldsymbol{\epsilon}_r = \alpha[10°, 10°, 10°]$ for rotation and $\boldsymbol{\epsilon}_t = \alpha[4, 4, 20]$ in meter for translation, where the scalar parameter $\alpha$ controls the overall tolerance level. Following the baseline (Anonymity), through a linear approximation, the corresponding keypoint error bounds can be derived from the pose error bounds, allowing the keypoint deviation constraints to also be expressed as a polytope. As $\alpha$ increases, the allowable keypoint deviations expand accordingly.

## 7.2 Metrics

We aim to answer two key questions:

1. How computationally efficient is the resulting neural network verification problem?

2. How accurate is the proposed certification method for robust pose estimation?

We evaluate performance using two metrics: verification time and verified rate. Verification time statistics—including the mean and standard deviation—are computed for each tested seed image under specific perturbations. The verified rate is defined as the proportion of cases in which the verification algorithm successfully certifies robustness at the keypoint level, relative to the total number of cases where the seed images yield acceptable keypoint errors.

**Empirical verified rate** Given a clean image and a specific type of perturbation, we apply a testing-based approach (Tian et al., 2018) to compute the empirical verified rate, which acts as an upper bound for the verified rate obtained through the verification method. Specifically, we randomly sample 100 instances within the given convex hull and run them through the neural network to check whether the keypoint prediction error for each instance satisfies the output specification. If all 100 instances meet the specification, the image is considered empirically robust to that perturbation.

## 7.3 Results

**Solver setup** We employ the verification toolbox $\texttt{ModelVerification.jl}$ (MV) (Wei et al., 2025), which accepts convex hulls as input specifications. $\texttt{MV.jl}$ is the state-of-the-art verifier that supports a wide range of verification algorithms and is the most user-friendly to extend. It follows a branch-and-bound strategy to divide and conquer the problem efficiently. Two parameters guide this process: $\texttt{split\_method}$ determines the division of an unknown branch into smaller branches for further refinement, and $\texttt{search\_method}$ dictates the approach to navigating through the branch. We set $\texttt{search\_method}$ to use breadth-first search and $\texttt{split\_method}$ to bisect the branch. The computing platform is a Linux server equipped with dual Intel

| $m$ | $\alpha = 1.0$ | | | $\alpha = 0.5$ | | | $\alpha = 0.2$ | | | $\alpha = 0.1$ | | |
|---|---|---|---|---|---|---|---|---|---|---|---|---|
| | testing | ours | baseline | testing | ours | baseline | testing | ours | baseline | testing | ours | baseline |
| 1 | 100.0% (200/200) | 99.5% (199/200) | 47.5% (95/200) | 100.0% (198/198) | 99.0% (196/198) | 0.0% (0/198) | 100.0% (190/190) | 71.6% (136/190) | 0.0% (0/190) | 88.7% (86/97) | 10.3% (10/97) | 0.0% (0/97) |
| 2 | 100.0% (200/200) | 99.0% (198/200) | 52.5% (105/200) | 100.0% (198/198) | 100.0% (198/198) | 0.0% (0/198) | 98.4% (187/190) | 72.1% (137/190) | 0.0% (0/190) | 89.7% (87/97) | 8.2% (8/97) | 0.0% (0/97) |
| 3 | 100.0% (200/200) | 99.0% (198/200) | 53.0% (106/200) | 100.0% (198/198) | 99.0% (196/198) | 0.0% (0/198) | 99.4% (189/190) | 67.4% (128/190) | 0.0% (0/190) | 88.7% (86/97) | 5.1% (5/97) | 0.0% (0/97) |
| 4 | 100.0% (200/200) | 97.0% (194/200) | 53.0% (106/200) | 100.0% (198/198) | 98.0% (194/198) | 0.0% (0/198) | 99.5% (189/190) | 62.1% (118/190) | 0.0% (0/190) | 90.7% (88/97) | 7.2% (7/97) | 0.0% (0/97) |

Table 1: Statistical results on verified rate for non-overlapping local perturbations, with "testing" indicating the empirical verified rate.

CPUs, each with 24 cores running at 2.20 GHz and a total of 376 GB of memory. Additionally, the server includes eight NVIDIA RTX A4000 GPUs, each with 16 GB of memory.

**Local object occlusions** The verified rates for non-overlapping images are presented in Table 1, with each cell displaying both the fraction and the corresponding percentage. Under the same keypoint error thresholds (i.e., fixed $\alpha$), our method consistently outperforms the baseline (Anonymity) in terms of verified rate, as shown by its smaller gap relative to the empirical verified rate obtained from testing. In contrast, the baseline (Anonymity) fails to verify any image when $\alpha \leq 0.5$, highlighting the conservativeness of their approach. Notably, our method maintains a relatively stable verified rate as the number of local objects ($m$) increases, suggesting that background perturbations have limited impact on model robustness. Moreover, for a fixed number $m$ of local objects, the verified rate using our method remains steady across $\alpha$ values from 1.0 to 0.5 and is close to the empirical verified rate, but drops significantly as $\alpha$ drops from 0.2 to 0.1. This indicates that tighter keypoint error bounds increase the influence of over-approximation, affecting verification performance under stricter accuracy requirements.

Similarly, Table 2 reports the verified rates for overlapping images. Unlike the non-overlapping case, for a given $\alpha$, all fractions share the same denominator, since they correspond to the same set of seed images yielding acceptable keypoint errors, regardless of the number of local perturbations ($m$). In this table, the denominator decreases as $m$ increases because the 20 objects were randomly placed on the images. Because the airplane occupies a relatively small portion of the image, these objects are more often disjoint from it, resulting in sufficient data for non-overlapping images across all values of $m$ from 1 to 4. In contrast, overlapping images require all samples to contain overlaps, leading to fewer available images as $m$ increases. Regarding verified rates, while our method continues to outperform the baseline (Anonymity) , both approaches achieve consistently lower verified rates compared to non-overlapping images. Furthermore, distinct from the non-overlapping case, our method's verified rate declines rapidly as the number of local objects ($m$) increases. This suggests that the neural network focuses heavily on the airplane, rendering it more vulnerable to perturbations directly affecting it.

The verification runtimes of both methods on non-overlapping images increase as $m$ grows, shown in Table 3, due to the expansion of the input set and corresponding reachable set. Interestingly, when holding $m$ constant, our method exhibits a decreasing trend in runtime as $\alpha$ decreases, while the baseline (Anonymity) shows the opposite. This leads to our method being slower at $\alpha = 1.0$, but faster at $\alpha = 0.1$. To explain this observation, Table 5 reports the MILP size statistics for our method when $m = 4$, including counts of binary, integer, and continuous variables, as well as constraints. For non-overlapping images, the MILP size remains relatively stable across different $\alpha$ values after applying pruning. The variation in runtime is instead attributed to MILP feasibility: smaller $\alpha$ leads to more feasible MILP problems, making it easier to quickly find a feasible solution, whereas larger $\alpha$ makes infeasibility proofs more computationally demanding. In contrast, the baseline (Anonymity) models each post-pooling pixel as a distinct class, so decreasing $\alpha$ increases the number of output categories, leading to higher output dimensionality and longer runtimes.

| $m$ | $\alpha = 1.0$ | | | $\alpha = 0.5$ | | | $\alpha = 0.2$ | | | $\alpha = 0.1$ | | |
|---|---|---|---|---|---|---|---|---|---|---|---|---|
| | testing | ours | baseline | testing | ours | baseline | testing | ours | baseline | testing | ours | baseline |
| 1 | 100.0% | 95.4% | 50.0% | 100.0% | 93.3% | 0.0% | 96.3% | 64.2% | 0.0% | 76.3% | 6.2% | 0.0% |
| | (196/196) | (187/196) | (98/196) | (195/195) | (182/195) | (0/195) | (183/190) | (122/190) | (0/190) | (74/97) | (6/97) | (0/97) |
| 2 | 100.0% | 77.1% | 41.1% | 100.0% | 78.9% | 0.0% | 96.7% | 43.6% | 0.0% | 79.1% | 3.3% | 0.0% |
| | (192/192) | (148/192) | (79/192) | (190/190) | (150/190) | (0/190) | (175/181) | (79/181) | (0/181) | (72/91) | (3/91) | (0/91) |
| 3 | 100.0% | 64.3% | 33.1% | 100.0% | 64.3% | 0.0% | 96.0% | 30.4% | 0.0% | 73.9% | 1.4% | 0.0% |
| | (157/157) | (101/157) | (52/157) | (157/157) | (101/157) | (0/157) | (142/148) | (45/148) | (0/148) | (51/69) | (1/69) | (0/69) |
| 4 | 100.0% | 51.6% | 31.0% | 100.0% | 46.8% | 0.0% | 89.0% | 20.9% | 0.0% | 67.9% | 1.9% | 0.0% |
| | (126/126) | (65/126) | (39/126) | (126/126) | (59/126) | (0/126) | (105/115) | (24/115) | (0/115) | (36/53) | (1/53) | (0/53) |

Table 2: Statistical results on verified rate for overlapping local perturbations, with "testing" indicating the empirical verified rate.

| $m$ | $\alpha = 1.0$ | | $\alpha = 0.5$ | | $\alpha = 0.2$ | | $\alpha = 0.1$ | |
|---|---|---|---|---|---|---|---|---|
| | ours | baseline | ours | baseline | ours | baseline | ours | baseline |
| 1 | 11.8±7.4 | 7.6±6.6 | 10.8±5.2 | 13.0±9.0 | 12.6±16.1 | 20.4±6.3 | 9.1±12.4 | 26.4±6.4 |
| 2 | 17.0±15.2 | 12.4±9.9 | 18.6±29.5 | 19.2±12.0 | 16.1±12.4 | 26.7±7.4 | 12.3±16.4 | 33.0±11.8 |
| 3 | 29.8±71.0 | 17.2±9.6 | 28.9±79.2 | 22.5±13.5 | 21.9±17.9 | 29.8±10.7 | 14.1±18.1 | 35.6±10.8 |
| 4 | 36.0±73.6 | 21.5±12.4 | 37.2±77.4 | 28.0±15.7 | 29.0±23.5 | 34.7±12.2 | 18.5±23.3 | 39.2±14.2 |

Table 3: Statistical results on verification time for non-overlapping local perturbations (seconds).

| $m$ | $\alpha = 1.0$ | | $\alpha = 0.5$ | | $\alpha = 0.2$ | | $\alpha = 0.1$ | |
|---|---|---|---|---|---|---|---|---|
| | ours | baseline | ours | baseline | ours | baseline | ours | baseline |
| 1 | 40.3±95.9 | 16.3±17.8 | 44.0±94.0 | 20.9±16.6 | 23.9±39.8 | 29.3±14.2 | 29.3±45.9 | 36.0±19.5 |
| 2 | 113.0±155.4 | 30.8±27.3 | 105.9±147.5 | 35.1±23.6 | 76.9±114.6 | 48.4±37.0 | 54.8±69.8 | 51.1±40.2 |
| 3 | 170.2±175.1 | 48.3±47.7 | 166.1±173.6 | 56.6±53.7 | 99.2±131.0 | 65.6±60.2 | 74.4±83.7 | 64.2±30.4 |
| 4 | 230.8±190.0 | 66.2±55.8 | 252.6±194.0 | 77.9±60.4 | 140.9±138.5 | 84.9±65.1 | 151.2±210.3 | 104.4±95.3 |

Table 4: Statistical results on verification time for overlapping local perturbations (seconds).

Table 4 shows that both methods experience increased verification times on overlapping images relative to non-overlapping ones, with our method becoming slower than the baseline. This behavior stems from the significantly larger and more complex reachable sets in the overlapping case, which reduce the effectiveness of the pruning strategy and lead to considerably larger MILP instances. As indicated in Table 5, although the original MILP sizes are comparable, pruning reduces the problem size by approximately three orders of magnitude for non-overlapping images, but only by about two orders for overlapping ones. Consequently, the pruned MILP for overlapping images remains roughly an order of magnitude larger than that for non-overlapping images.

**Global perturbations** A similar pattern is seen for brightness and contrast perturbations, as shown in Tables 6-9. Interestingly, the verified rates remain similar across different types of perturbations for the same $\alpha$ values, whether the perturbations are non-overlapping local occlusions or global changes like brightness and contrast. This indicates that the neural network exhibits a certain level of general robustness to diverse types of perturbations with moderate intensity.

# 8 Discussion and Conclusion

In this work, we propose a verification method that combines reachability analysis with MILP for learning-based keypoint detection, allowing the output specification to capture interdependencies among keypoints.

| $\alpha$ | overlapping | pruning | binary | integer | continuous | constraints |
|---|---|---|---|---|---|---|
| 1.0 | ✗ | ✗ | $9.42\times10^4\pm0.00\times10^1$ | $46.0\pm0.0$ | $1.89\times10^5\pm5.29\times10^2$ | $6.61\times10^5\pm1.06\times10^3$ |
| | | ✓ | $5.31\times10^1\pm4.29\times10^1$ | $46.0\pm0.0$ | $8.36\times10^2\pm7.48\times10^2$ | $1.95\times10^3\pm1.61\times10^3$ |
| | ✓ | ✗ | $9.42\times10^4\pm0.00\times10^1$ | $46.0\pm0.0$ | $1.92\times10^5\pm6.64\times10^3$ | $6.67\times10^5\pm1.33\times10^4$ |
| | | ✓ | $8.18\times10^2\pm4.11\times10^3$ | $46.0\pm0.0$ | $5.07\times10^3\pm1.42\times10^4$ | $1.27\times10^4\pm4.06\times10^4$ |
| 0.5 | ✗ | ✗ | $9.42\times10^4\pm0.00\times10^1$ | $46.0\pm0.0$ | $1.89\times10^5\pm4.41\times10^2$ | $6.61\times10^5\pm8.97\times10^2$ |
| | | ✓ | $4.64\times10^1\pm1.60\times10^1$ | $46.0\pm0.0$ | $7.47\times10^2\pm4.93\times10^2$ | $1.75\times10^3\pm1.03\times10^3$ |
| | ✓ | ✗ | $9.42\times10^4\pm0.00\times10^1$ | $46.0\pm0.0$ | $1.92\times10^5\pm6.78\times10^3$ | $6.67\times10^5\pm1.35\times10^4$ |
| | | ✓ | $1.03\times10^3\pm4.98\times10^3$ | $46.0\pm0.0$ | $6.04\times10^3\pm1.74\times10^4$ | $1.53\times10^4\pm4.96\times10^4$ |
| 0.2 | ✗ | ✗ | $9.42\times10^4\pm0.00\times10^1$ | $46.0\pm0.0$ | $1.89\times10^5\pm5.08\times10^2$ | $6.60\times10^5\pm1.02\times10^3$ |
| | | ✓ | $4.78\times10^1\pm2.09\times10^1$ | $46.0\pm0.0$ | $7.89\times10^2\pm5.37\times10^2$ | $1.84\times10^3\pm1.13\times10^3$ |
| | ✓ | ✗ | $9.42\times10^4\pm0.00\times10^1$ | $46.0\pm0.0$ | $1.92\times10^5\pm4.70\times10^3$ | $6.66\times10^5\pm9.43\times10^3$ |
| | | ✓ | $9.45\times10^2\pm3.10\times10^3$ | $46.0\pm0.0$ | $5.77\times10^3\pm1.14\times10^4$ | $1.45\times10^4\pm3.19\times10^4$ |
| 0.1 | ✗ | ✗ | $9.42\times10^4\pm0.00\times10^1$ | $46.0\pm0.0$ | $1.89\times10^5\pm5.01\times10^2$ | $6.60\times10^5\pm1.04\times10^3$ |
| | | ✓ | $4.90\times10^1\pm4.09\times10^1$ | $46.0\pm0.0$ | $7.64\times10^2\pm6.75\times10^2$ | $1.79\times10^3\pm1.46\times10^3$ |
| | ✓ | ✗ | $9.42\times10^4\pm0.00\times10^1$ | $46.0\pm0.0$ | $1.92\times10^5\pm6.33\times10^3$ | $6.67\times10^5\pm1.27\times10^4$ |
| | | ✓ | $1.95\times10^3\pm7.19\times10^3$ | $46.0\pm0.0$ | $9.07\times10^3\pm2.49\times10^4$ | $2.41\times10^4\pm7.12\times10^4$ |

Table 5: Statistical results of the MILP size, including the number of binary, integer, and continuous variables, as well as the number of constraints. These results vary with the pose error threshold $\alpha$, the presence of object-airplane overlap, and the use of the pruning strategy. Results obtained with pruning are highlighted in gray for non-overlapping images and in orange for overlapping images.

| $c\%$ | $\alpha = 1.0$ | | | $\alpha = 0.5$ | | | $\alpha = 0.2$ | | | $\alpha = 0.1$ | | |
|---|---|---|---|---|---|---|---|---|---|---|---|---|
| | testing | ours | baseline | testing | ours | baseline | testing | ours | baseline | testing | ours | baseline |
| 0.05 | 100.0% | 99.5% | 55.5% | 100.0% | 100.0% | 0.0% | 100.0% | 75.3% | 0.0% | 94.8% | 11.3% | 0.0% |
| | (200/200) | (199/200) | (111/200) | (198/198) | (198/198) | (0/198) | (190/190) | (143/190) | (0/190) | (92/97) | (11/97) | (0/97) |
| 0.5 | 100.0% | 99.5% | 55.0% | 100.0% | 100.0% | 0.0% | 99.5% | 72.1% | 0.0% | 91.8% | 10.3% | 0.0% |
| | (200/200) | (199/200) | (110/200) | (198/198) | (198/198) | (0/198) | (189/190) | (137/190) | (0/190) | (89/97) | (10/97) | (0/97) |
| 1 | 100.0% | 99.5% | 54.0% | 100.0% | 100.0% | 0.0% | 98.4% | 64.7% | 0.0% | 84.5% | 8.2% | 0.0% |
| | (200/200) | (199/200) | (108/200) | (198/198) | (198/198) | (0/198) | (187/190) | (123/190) | (0/190) | (82/97) | (8/97) | (0/97) |

Table 6: Statistical results on verified rate for contrast.

| $c\%$ | $\alpha = 1.0$ | | $\alpha = 0.5$ | | $\alpha = 0.2$ | | $\alpha = 0.1$ | |
|---|---|---|---|---|---|---|---|---|
| | ours | baseline | ours | baseline | ours | baseline | ours | baseline |
| 0.05 | 19.0±2.8 | 16.1±3.4 | 19.4±2.7 | 123.3±9.6 | 20.1±3.6 | 29.1±3.7 | 23.9±3.5 | 34.7±3.2 |
| 0.5 | 22.9±2.9 | 19.3±3.4 | 23.3±2.9 | 26.9±9.3 | 24.1±3.7 | 33.0±3.8 | 28.0±3.4 | 38.9±3.0 |
| 1 | 48.1±9.4 | 45.0±10.3 | 49.0±9.7 | 53.3±16.0 | 49.7±9.8 | 58.9±9.9 | 54.9±9.8 | 65.6±8.8 |

Table 7: Statistical results on verification time for contrast (seconds).

Our approach encodes the coupling between keypoints directly, without decoupling, and is applicable to general heatmap-based keypoint detection models. Experimental results show that our coupled method achieves significantly higher verified rates compared to the decoupled approach.

A key limitation of the current framework is that, under strict output specifications, there remains a noticeable gap between the verified robustness rates and the empirical robustness observed through testing-based evaluation. This gap stems from the conservativeness of reachable set over-approximations. Future work will

| $b$ | $\alpha = 1.0$ | | | $\alpha = 0.5$ | | | $\alpha = 0.2$ | | | $\alpha = 0.1$ | | |
|---|---|---|---|---|---|---|---|---|---|---|---|---|
| | testing | ours | baseline | testing | ours | baseline | testing | ours | baseline | testing | ours | baseline |
| 1 | 100.0% (200/200) | 99.5% (199/200) | 55.5% (111/200) | 100.0% (198/198) | 100.0% (198/198) | 0.0% (0/198) | 100.0% (190/190) | 72.1% (137/190) | 0.0% (0/190) | 93.8% (91/97) | 10.3% (10/97) | 0.0% (0/97) |
| 2 | 100.0% (200/200) | 99.0% (198/200) | 54.5% (109/200) | 100.0% (198/198) | 100.0% (198/198) | 0.0% (0/198) | 100.0% (190/190) | 63.7% (121/190) | 0.0% (0/190) | 88.6% (86/97) | 7.2% (7/97) | 0.0% (0/97) |

Table 8: Statistical results on verified rate for brightness.

| $b$ | $\alpha = 1.0$ | | $\alpha = 0.5$ | | $\alpha = 0.2$ | | $\alpha = 0.1$ | |
|---|---|---|---|---|---|---|---|---|
| | ours | baseline | ours | baseline | ours | baseline | ours | baseline |
| 1 | 33.4±5.5 | 29.0±5.6 | 33.9±5.5 | 35.2±9.8 | 34.3±5.9 | 43.4±6.4 | 39.6±11.9 | 48.9±5.4 |
| 2 | 65.6±13.4 | 58.3±11.2 | 67.6±16.9 | 65.0±15.1 | 66.2±13.3 | 75.8±11.6 | 72.1±12.8 | 77.9±11.0 |

Table 9: Statistical results on verification time for brightness (seconds).

focus on developing tighter reachability approximations and scalable verification strategies that can further reduce this gap and extend the applicability of the proposed method to larger and more complex keypoint detection networks.

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
