# OpenReview forum: "From Decoupled to Coupled: Robustness Verification for Learning-based Keypoint Detection with Joint Specifications"
_TMLR — Rejected by TMLR_

### Review · Reviewer_jufL · 2025-10-22

**Summary Of Contributions:**

The authors study formal verification of neural networks, applied to the task of heatmap-based keypoint detection in images. They propose a verification method combining reachability analysis with a mixed-integer linear program (MILP). The proposed method is evaluated using a small CNN and a dataset of ~7k images of airplanes. Its performance in terms of verification time and verified rate is compared to a baseline method from an anonymous paper, achieving higher verified rates.



***
Strengths:
- The paper is well written overall in the sense that it contains virtually no typos or similar issues.
- The studied problem is quite interesting and important, applying formal verification to a more challenging and practically relevant task such as image keypoint detection seems quite rare in previous work.
- The verified rate is consistently improved compared to the baseline method.





***
Weaknesses:
- The performance is only compared to a single baseline method (which also is completely anonymous, difficult to determine how strong of a baseline this actually is).
- At least personally, I found Section 5 difficult to follow and understand. Moreover, the general "flow" of the paper could be improved, making the transitions from Section 4 --> 5 --> 6 --> 7 more smooth.
- The practical implications of the experimental results are not properly discussed. In Section 8 the authors write that _``our coupled method achieves significantly higher verified rates compared to the decoupled approach''_, yet _``given strict output specifications, there remains a substantial gap between the verified rates achieved by our coupled method and the empirical rates obtained through the testing-based method''_. So, what is the main practical takeaway from this then? How good is the performance of the proposed method in an _absolute_ sense (yes, it's good relative to the baseline, but does this actually work?)? Does it work well enough to actually be used in practice? In what settings?




***
Questions/suggestions:
- I quite like the start of Section 5, but then the transition to 5.1 could be more smooth. What are you doing in 5.1 and the remaining subsections? Why? What is it that you're building towards in this section? I think you could guide the reader through Section 5 better, make all subsection transitions more smooth.
- At least personally, I found 5.3 particularly difficult to follow/understand. I appreciate that you give an example, but this was _a lot_ of details, at least for me.
- The transition from Section 5 to Section 6 is also a bit odd, having Section 6 solely consist of Theorem 1, without any comments or discussion.
- I would definitely like at least a short introduction paragraph to be added to the start of Section 7 (what will you be doing in the different subsections of 7? Guide the reader a bit).
- At least some of the results in Table 1 - 9 could perhaps be presented in some other, more illustrative way, instead of just having large tables with a lot of numbers?
- I think a Discussion section should be added after Section 7, before the summarizing conclusion.
- In general, the practical utility of the proposed method (and other formal verification methods, _I'm not overly familiar with this area_) is somewhat unclear to me. Robustness is always checked against a specific set of image perturbations, e.g. brightness/contrast within a certain range. But does this actually help in practice, in real-world applications, where input images can be basically anything?





***
Minor things:
- Figure 1 caption: "whether the model is robustness" --> "whether the model is robust"?
- 2.1: "multi-linear perceptrons (MLPs)" --> "multi-layer perceptrons (MLPs)"?
- 3.1: "it’s often essential" --> "it is often essential"?
- 3.2, "NN verification", perhaps just define "NN" prior to this?
- Final sentence of 7.1.3, missing "{"?

**Additional Comments:**

No additional comments.

**Audience:**

Yes

**Audience Explanation:**

The studied problem is quite interesting and important.

**Broader Impact Concerns:**

No concerns.

**Claims And Evidence:**

No

**Claims Explanation:**

The method description could be improved, and the practical implications of the results are not properly discussed.

**Requested Changes:**

This is a quite well-written paper overall that I think could be relevant for the TMLR audience.

However, I think the current version requires a number of clarifications and modifications, see "Weaknesses" and "Questions/suggestions" above.

---

### Review · Reviewer_HiWy · 2025-10-23

**Summary Of Contributions:**

**[A general summary]** This paper tackles the underexplored problem of formal robustness verification for neural keypoint detectors. Previous methods simplify verification by decoupling each keypoint as an independent classification problem, ignoring inter-keypoint dependencies that are critical for downstream tasks. The authors introduce a coupled robustness verification framework for heatmap-based keypoint detectors that directly encodes joint specifications among keypoints. Their framework formulates robustness verification as a Mixed-Integer Linear Program (MILP) that integrates. It includes a reachable set of heatmaps from a neural backbone and a polytope constraint that specifies allowable joint deviations among keypoints. If the MILP is infeasible, robustness is certified (sound guarantee); if feasible, it yields potential counterexamples. The method is evaluated on an airplane-pose dataset, showing higher verified rates than previous (decoupled) approaches, especially under tight keypoint error bounds.

**[Strengths]**
- First formal coupled verification framework for keypoint detectors, generalizable to any heatmap-based architecture.
- The paper proves soundness of the MILP-based certification (Theorem 1)
- Comprehensive experiments on occlusion and brightness/contrast perturbations, demonstrating tangible improvements in verified rate over prior methods.
- Integrates formal methods (reachability + MILP) with vision tasks that are often treated empirically.

**[Weakness]**
- MILP formulation can become large, especially for overlapping perturbations or stricter constraints; runtimes can reach hundreds of seconds.
- The verification may fail when reachable sets are over-approximated.
- Only one dataset (airplane pose estimation) and a relatively small network (0.65M parameters) are used; no experiments on standard HPE datasets (COCO, etc.).
- No demonstration of how such formal verification scales to real-world models or diverse architectures.

**Additional Comments:**

The paper presents an interesting and technically solid framework for coupled robustness verification in keypoint detection. The idea is novel and well-motivated, and the results are promising. However, the main drawback is that validation on only a single dataset limits confidence in the method’s practical applicability. Expanding experiments to additional datasets of other keypoints-based vision task would strengthen the paper. The authors may also consider shortening some sections or paragraphs to improve readability and conciseness.

**Audience:**

Yes

**Audience Explanation:**

This paper will interest a specific but meaningful subset of the TMLR community, particularly those focused on safety-critical perception, and formal methods for deep vision systems.

**Broader Impact Concerns:**

No major ethical issues are apparent. The work aims to improve safety and reliability of perception systems.

**Claims And Evidence:**

Yes

**Claims Explanation:**

Yes, the main claims are mostly supported by accurate and convincing evidence, though scalability and generalization evidence remain limited.

**Requested Changes:**

- A broaden experimental validation to other standard keypoints benchmarks (e.g., COCO, MPII).  Current single-dataset evaluation truly limits generalizability, espeically author mentioned human pose and 3D view points in the abstract.

- Include runtime and accuracy trade-off analysis across larger models. This would clarify scalability and practical feasibility.

- Clarify limitations of the sound-only guarantee. It should help readers understand that unverified ≠ non-robust, improving interpretability.

---

### Review · Reviewer_UsDW · 2025-10-29

**Summary Of Contributions:**

This paper presents robustness verification framework for heatmap-based keypoint detectors that preserves interdependencies among keypoints. Key innovations include:
- A MILP formulation for robustness verification for keypoint detection with joint deviation constraints.
- General approach that potentially supports different keypoint deteciton backbone
- Dynamic indexing mechanism for variable keypoint positions in verification
- Theoretical  guarantees and practical pruning strategies for the MILP problem.

Authors validate their approach on a small scale dataset comprising 7000 images and show improvement over decoupled baselines in verified rates.

**Additional Comments:**

- It would be nice to add  a discussion to describe why generated pertubation are representative of case that can applied in practice
- Could you explain more the tradeoff with some of the assumption made in the theoretical part, such as the zonotope reachable set?

**Audience:**

Yes

**Audience Explanation:**

I think evaluating robustness of keypoint detector could be detector is a relevant topic for TMLR. Yet, I think the study should go beyond one dataset and the specific case of airplane to improve its significance.

**Broader Impact Concerns:**

No concern

**Claims And Evidence:**

No

**Claims Explanation:**

While the approach appears theoretically sound, several improvements can be made to the empirical study to better validate the claims:
- The authors only validate their approach on a small-scale dataset comprised of only 7,000 images of airplanes. It's unclear whether the results would remain true for larger-scale datasets or for other domains (e.g., face or body keypoint detection).
- One of the claims is the generality of the approach to many keypoint-detection backbones, yet the authors only evaluate their approach on one backbone in their empirical section.
- Robustness is verified with respect to artificial perturbations defined by the authors. Do these artificial perturbations correlate with real-world perturbations or out-of-distribution examples?
- It's unclear what the performance of the keypoint detector is in terms of keypoint accuracy and whether the approach could be applied to state-of-the-art methods.

**Requested Changes:**

- Show empirically that the approach can be applied to larger dataset and broader visual domain.
- Show empirically that the approach can be leveraged for different keypoint backbone.
- Investigate the accuracy of keypoint detectors and show that the approach can be applied to state-of-art keypoint detectors.

---

### Decision · Action_Editor_36e2 · 2026-01-20

**Recommendation:** Reject

**Audience:**

Yes

**Audience Explanation:**

Topically this paper fits well within the scope of TMLR's readership.

**Claims And Evidence:**

No

**Claims Explanation:**

The authors propose a formal verification framework for heatmap-based keypoint detection in images. The proposed method combines reachability analysis with a mixed-integer linear program (MILP). The method is evaluated using a small CNN backbone on a small dataset for testing keypoint detection on airplane images.

The paper received mixed reviews. The reviewers praised the uniqueness of the approach and the importance of integrating formal verification into keypoint detection tasks, while raising the computational challenges of verification as network sizes increase. All reviewers raised the issue that the method was only evaluated on a small dataset, using only a single backbone, and compared to only a single, anonymous baseline.

I agree with the reviewers on this aspect. The paper makes the claim that the approach is applicable to general models, and achieves significantly higher verified rates. However, to be published at TMLR, this claim needs to be backed up by more convincing evidence, either in terms of more experiments on more challenging (and state-of-the-art) image datasets, or in terms of a broader diversity of models, or ideally both. Moreover, the computational tractability issue (in my opinion) isn't fully addressed and the paper should have a discussion of potential limitations of the approach. I encourage the authors to take both these points into account while considering future revisions.

**Resubmission Of Major Revision:**

The authors may consider submitting a major revision at a later time.